# Surgical Management of Gastroenteropancreatic Neuroendocrine Tumors

**DOI:** 10.3390/cancers17030377

**Published:** 2025-01-23

**Authors:** Lisa M. Kenney, Marybeth Hughes

**Affiliations:** 1Department of Surgery, Eastern Virginia Medical School, Macon and Joan Brock Virginia Health Sciences at Old Dominion University, 825 Fairfax Avenue, Suite 610, Norfolk, VA 23507, USA; lisa.kenney.md@gmail.com; 2Department of Surgery, Division of Surgical Oncology, Eastern Virginia Medical School, Macon and Joan Brock Virginia Health Sciences at Old Dominion University, 825 Fairfax Avenue, Suite 610, Norfolk, VA 23507, USA

**Keywords:** neuroendocrine tumors, gastroenteropancreatic, surgical indications, metastasectomy

## Abstract

This review article discusses the surgical management of neuroendocrine tumors (NETs), a heterogeneous group of malignancies that often originate in the gastroenteropancreatic tract. Clinical outcomes in patients with NETs depend on tumor grade, size, and metastatic stage. Tailored surgical strategies such as endoscopic or formal surgical resection vary based on tumor grade and organ of origin, and can play a curative role in localized diseases. Additionally, cytoreduction surgery, liver-directed therapies, or liver transplants are options for symptom management and underscore the importance of a multidisciplinary approach.

## 1. Introduction

Neuroendocrine tumors (NETs) are a heterogeneous group of malignancies that arise from enterochromaffin cells and frequently originate in the gastrointestinal (GI) tract and retroperitoneum, including the stomach, small intestine, colon and rectum, appendix, and pancreas [1]. There has been an observed 6-fold increase in the incidence of NETs in the last four decades in the United States [2], and they occur in an estimated 8.19 per 100,000 people annually [3], with the most significant increase observed in grade 1, localized-stage, and appendix NETs [3]. According to the Surveillance, Epidemiology, and End Results (SEER) database, at the time of diagnosis, 53% of patients with NETs present with localized disease, 20% have locoregional disease, and 27% have distant metastases [4]. The median overall survival (OS) for all NET patients has been reported to be 9.3 years, with well-differentiated tumors being associated with a higher OS than moderately differentiated tumors (16.2 vs. 8.3 years). The poorly differentiated, undifferentiated, and anaplastic tumors have a reported OS of 10 months [4].

NETs are classified as either functional or non-functional, depending on their ability to secrete active hormones. The clinical heterogenicity of NETs are explained by their different genomic characteristics, and molecular findings underly the different grades and primary sites of origin [5]. NETs arise from a combination of genetic, epigenetic, and signaling pathway alterations. The mTOR, PI3K/Akt, and Notch pathways, along with angiogenic factors like vascular endothelial growth factor (VEGF), play a crucial role in tumor growth and proliferation. Key genetic mutations in NET development include those in MEN1, VHL, PTEN, DAXX, TSC1/TSC2, and ATRX. Sporadic mutations in RB1 and TP53 genes are associated with poorly differentiated NETs. Epigenetic changes, such as DNA methylation and histone modifications, and microRNA dysregulation, further contribute to tumor development [6,7,8].

Systemic treatment options for NETs allow symptomatic control. Examples of treatments and targeting pathways include everolimus, which targets the mTOR signaling pathway [9,10], and sunitinib, which has antiangiogenic function by targeting VEGF and the platelet-derived growth factor (PDGF) [11,12]. Temozolomide and capecitabine serve as chemotherapy options by acting as alkylating agents that induce DNA methylation and inhibit thymidylate synthase [13]. Somatostatin functions to control both tumor size and function by dampening the insulin-like growth factor receptor PI3K/Akt axis [6,14]. Peptide receptor radionuclide therapy with lutetium-177 dotatate (^177^Lu-DOTATATE) is also an option for advanced gastroenteropancreatic NETs [15].

While treatment of GI NETs involves a multidisciplinary approach, surgical resection of NETs is the cornerstone of treatment and the primary curative option, particularly for localized and well-differentiated tumors that may be curative in the case of R0 resection. In cases of metastatic disease, surgery is an option for symptom control and for the reduction in tumor burden [16]. The surgical approach for GI NETs depends on the location, size, grade, and metastatic potential of the tumor. This review discusses the surgical management of NETs that arise from the pancreas or GI tract.

## 2. Surgical Management of Gastric NETs

Gastric NETs are categorized into three types by the Rindi Classification and are also graded into three tumor classifications by the World Health Organization (WHO). Grade 1 (G1) tumors have a Ki67 proliferation index of <3% and mitotic index per high power field of <2 mitoses/2 mm^2^, Grade 2 (G2) tumors have a Ki67 proliferation index of 3–20% and mitotic index per high power field of 2–20 mitoses/2 mm^2^, and Grade 3 (G3) tumors have a Ki67 proliferation index of >20% and mitotic index per high power field of >20 mitoses/2 mm^2^ [17,18]. Neuroendocrine carcinomas (NECs) are additionally defined by WHO by having high-grade features and are subclassified as small- or large-cell types (SmCNEC, LCNEC) [18] (Table 1, Figure 1).

Per the Rindi Classification, Type I gastric NETs are associated with chronic autoimmune atrophic gastritis and tend to present as multiple sub-centimeter lesions that develop because of chronic G cell stimulation due to achlorhydria and intrinsic factor deficiency. Type I gastric NETs account for approximately 50–70% of gastric NETs, are typically well-differentiated lesions, and tend to be confined to the mucosa or submucosa layers of the stomach [20,21]. Type II gastric NETs, which account for approximately 5% or less of gastric NETs, are associated with multiple endocrine neoplasia type 1 (MEN1) Zollinger–Ellison syndrome gastrinomas. They also typically present with multiple sub-centimeter well-differentiated lesions. Although they have increased metastatic potential, metastasis only occurs in approximately 12% of these lesions [20]. Treatment of the underlying gastrinoma may cause regression of Type II gastric NETs [22], and reduce the need for further gastric surgery.

Surgical management for both Type I and Type II and gastric NETs is approached as such: if the lesion is <1 cm, it can be observed or resected endoscopically assuming there is no muscularis propria (MP) invasion or lymph node involvement. According to National Comprehensive Cancer Network (NCCN) guidelines, both Type I and Type II lesions >1 cm can be managed with endoscopic resection of lesions if they are well-differentiated G1 or G2 tumors [23]. North American Neuroendocrine Tumor Society (NANETS) also recommends endoscopic resection for Type I and Type 2 gastric NETs < 2 cm [24]. However, according to European Neuroendocrine Tumor Society (ENETS) guidance, Type II lesions should instead be treated with surgical resection due to the higher risk of metastasis [25]. In the case of MP invasion, lymphovascular invasion, or if the lesions are larger than 2 cm, per the ENETS and NANETS guidelines, partial or total gastrectomy with regional lymphadenectomy should be performed [24,26]. Japanese Neuroendocrine Tumor Society (JNETS) guidelines recommend a D2 dissection in this case [27]. Type I and Type II gastric NETs that measure from 1 to 2 cm can be managed with either endoscopic resection or gastrectomy and lymphadenectomy, with clinical decision-making made on an individual basis [26]. Type III gastric NETs, also known as sporadic gastric NETs, account for 15% of gastric NETs and tend to be solitary lesions that present with a gastric ulcer. They are not associated with syndromic conditions or with the overproduction of gastrin. The malignant potential of Type III gastric NETs is much higher than Type I and Type II gastric NETs, with most showing lymphovascular invasion at the time of diagnosis. Metastasis is found in 50–70% of well-differentiated Type III lesions and up to 100% of poorly differentiated tumors [28,29,30,31]. Though some have advocated for endoscopic resection as a first treatment in lesions < 2 cm that are well differentiated with no evidence of lymphovascular invasion [32,33], including in recent ENETS guidelines [26], radical resection with gastrectomy and lymphadenectomy remains the standard practice for Type III gastric NETs due to their high malignant potential [34]. Limited wedge resection with local nodal sampling (without standard lymphadenectomy) can be considered as a treatment option in patients with localized, G1–G2 Type III gastric NETs, with no evidence of lymphadenopathy on full staging preoperative imaging [26] (Table 2).

## 3. Surgical Management of Small Bowel NETs

Duodenal and periampullary NETs account for 11% of small bowel NETs [35] and are commonly located in the first or second part of the duodenum [36,37]. The WHO classification divides NETs of the duodenum and upper jejunum into three categories, with Class Ia being well-differentiated tumors that are non-functioning, confined to the mucosa or submucosa, no lymphovascular invasion, and limited to 1 cm or less in size. Class Ib tumors are also confined to the mucosa or submucosa but may have angioinvasion and may be >1 cm in size. Class Ib NETs can include functioning gastrinoma lesions that are sporadic or associated with MEN1, or non-functioning somatostatin-producing NETs or non-functioning serotonin-producing tumors. Class Ia and Ib tumors account for anywhere from 50 to 75% of duodenal and upper jejunum NETs. Class 2 NETs of the duodenum and upper jejunum are well-differentiated neuroendocrine carcinomas, have invasion of the muscularis propria or metastases, and account for 25–50% of duodenal or upper jejunum NETs. Class 3 lesions are classified as poorly differentiated neuroendocrine carcinomas and account for 1–3% of duodenal or upper jejunum NETs [18]. Class 3 lesions tend to occur in the periampullary region [33], and periampullary NETs have a higher incidence of poorly differentiated lesions than duodenal NETs in other areas [38]. Duodenal NETs also have five histologic subtypes including gastrinomas, somatostatinomas, non-functioning tumors, poorly differentiated neuroendocrine carcinomas, and gangliocytic paragangliomas [38]. A total of 90% of all duodenal NETs are non-functional. A total of 10% of patients with duodenal NETs have symptoms related to Zollinger–Ellison syndrome, and carcinoid syndrome is present in 3% of those with duodenal NETs [25]. While duodenal NETs tend to be solitary, multiple duodenal gastrinomas are usually found in the case of Zollinger–Ellison syndrome [36]. Although most duodenal NETs are limited to the mucosa or submucosa and measure < 2 cm in size [36,38], about half of these lesions are associated with lymph node metastasis, and 10% have distant metastasis [39].

Surgical management for duodenal NETs depends on the tumor size, location, and histology. For lesions 1 cm or smaller in non-periampullary locations without metastasis or hormonal syndromes, these NETs can be resected by endoscopic techniques. However, if NETs of any size are present in the periampullary region, surgical resection and lymphadenectomy is recommended due to the high risk of poorly differentiated lesions, increased risk of metastasis, and poorer survival associated with these lesions [40,41,42]. While some advocate that low-grade (G1 or G2) periampullary NETs can be resected with endoscopic techniques, prospective multi-center studies are currently not available to support this approach [43]. NETs in the duodenum measuring 2 cm or larger or any size with evidence of lymph node metastasis should be treated with surgical resection. For non-periampullary lesions that measure between 1 and 2 cm, there is no standard technique for surgical excision (endoscopic vs. surgical resection), and should be managed on an individual patient basis [44]. If tumors are endoscopically resected, follow-up with endoscopic surveillance is recommended.

Small bowel NETs involving the jejunum or ileum are often not diagnosed until they cause obstruction, abdominal pain, carcinoid syndrome from metastatic disease, or, rarely, episodes of mesenteric ischemia [45,46]. As a result, an estimated 35–60% of patients who present with small bowel NETs have metastatic disease at the time of diagnosis, and the reported rate of lymph node metastasis in patients with small bowel NETs ranges from 46 to 98% [47,48,49,50,51]. However, despite its advanced presentation, the median survival of patients with metastatic small bowel NETs is 56 months, and even further improved survival has been associated with lymphadenectomy and cytoreduction surgery [47,52,53]. The standard technique for surgical resection of small bowel NETs is resection of primary tumors, regional lymph nodes, and the associated mesentery, and resection of any mesenteric and peritoneal masses with grossly negative margins. This may also include resection or ablation of hepatic metastases. Palpation of the entire length of the small bowel is the most sensitive method for identifying multifocal tumors, which has historically been described to be present in 20% of cases [54,55,56], but in more recent series, has been described to be present in 50% of patients [57,58]. Thus, open mini-laparotomy or hand-assisted surgical resection is the preferable approach to intra-corporeal surgery alone. Although new series are starting to incorporate these techniques, long-term outcomes are lacking (Table 3).

## 4. Surgical Management of Pancreas NETs

Pancreatic NETs account for approximately 2% of all pancreatic neoplasms [59]. Like other NETs, they are categorized as functional or non-functional depending on their ability to produce biologically active hormones. The primary prognostic indicator for pancreatic NET is tumor grade, defined by the Ki67 proliferation index and the mitotic index by the WHO, as described previously [17]. Poorly differentiated pancreatic NETs are divided into two classifications, small cell and large cell, both having a Ki67 proliferation index of >20% and mitotic index per high power field of >20 mitoses/2 mm^2^. While histologic grade is the most important factor for the prognosis of pancreatic NETs, for lesions < 2 cm, diagnostic yield from endoscopic ultrasound with fine-needle aspiration (FNA) is poor, with differentiation and Ki-67 index determined in only 26.4% and 20.1% of patients, respectively [60].

Functional pancreatic NETs that are localized are generally managed with surgical resection for the purposes of symptom control and cure, with improved survival for all stages of disease following resection [61] and should be considered regardless of the size of the lesion. Insulinomas have a 10% chance of malignancy, and as such, are generally managed with enucleation or pancreas-sparing procedures without lymphadenectomy [62]. Enucleation is an option when the lesion is superficial or embedded in the parenchyma at least 2 mm from the main pancreatic duct or portal vein, to avoid pancreatic fistula. When lesions do not meet the criteria for enucleation, surgical resection includes distal pancreatectomy for distal lesions and pancreaticoduodenectomy for lesions in the pancreatic head [63]. While laparoscopic enucleation has a lower morbidity than an open approach [50], enucleation has a higher risk of pancreatic leak than distal pancreatectomy or pancreaticoduodenectomy [64]. In selected cases, particularly for small, localized insulinomas (<2 cm) in patients who are poor surgical candidates, endoscopic ultrasound-guided radiofrequency has emerged as an alternative to surgical resection. This approach delivers thermal energy to ablate the tumor under ultrasound guidance. Early studies demonstrate effective local tumor control and symptomatic relief. However, long-term survival and recurrence data remain under investigation [65,66]. Gastrinomas have an estimated 70% incidence of malignancy [67], and as such, the standard surgical excision involves resection of the involved pancreas along with routine lymph node dissection of peripancreatic, periduodenal, and pancreaticoduodenal areas [68,69]. Gastrinomas commonly present in the area are referred to as the gastrinoma triangle (See Figure 2), which is the area between the junction of the cystic duct and common bile duct, the body and neck of the pancreas, and the second and third portions of the duodenum [70,71]. In cases of surgical resection, operative exploration of this area has been suggested to identify these lesions as they can be multifocal, though with modern imaging techniques, this is not as necessary [72]. In the case of MEN1, there is no current international consensus on the management of gastrinomas given the high rate of recurrence, lack of benefit from incomplete resection, and the often slow-growing nature of the disease [73], with the NCCN recommending observation and treatment with proton pump inhibitors or exploratory surgery for occult gastrinomas, and JNETS guidelines recommending resection of all functioning pancreatic NETs, including gastrinomas associated with MEN1 [74]. Many centers approach MEN1-related gastrinomas with a non-surgical management approach unless lesions reach 2 cm or larger, with disease-related survival improved with surgery in this population [75,76]. The majority of glucagonomas and VIPomas are malignant and, as such, will also involve a distal pancreatectomy or pancreaticoduodenectomy procedure for resection, along with lymph node excision when presenting as resectable [77]. Although somatostatinomas have a high risk of malignancy (78%), with 70–92% presenting with metastatic disease [78,79], localized lesions smaller than 2 cm may be resected with either formal resection or enucleation. Since somatostatinomas tend to arise in the pancreatic head, a formal resection would involve a pancreaticoduodenectomy [78,80].

Most pancreatic NETs are non-functional, with 60–90% of pancreatic NETs not secreting any active hormones [81]. Due to their common asymptomatic progression, they can often present with metastatic disease [82]. The surgical goal in the case of non-functional pancreatic NETs is to prevent metastatic spread and improve survival. Studies have demonstrated a direct relationship between tumor size and risk of metastasis [83], with only 6% of non-functional pancreatic NETs less than 2 cm in size being metastatic at diagnosis. Given this low risk of metastasis, it is acceptable practice to observe lesions smaller than 2 cm [84,85,86]. However, some have advocated for a more aggressive approach, noting a metastatic rate of 8% in tumors as small as 1.5 cm [87]. When surgical excision is performed, enucleation is an acceptable option for small lesions. However, formal anatomic resection (pancreaticoduodenectomy or distal pancreatectomy) is recommended with regional lymphadenectomy for lesions > 2 cm in size. Since non-functioning pancreatic NETs associated with MEN1 tend to be multifocal and have a low risk of progression, surgery is generally reserved for lesions greater than 1–2 cm in size with growth [77] (Table 4).

## 5. Surgical Management of Colorectal NETs

The most common location of NETs in the colon is in the ascending colon. While there are no current consensus guidelines on the surgical management of these lesions, a comparative study of ileocecectomy and right hemicolectomy for ileocecal NETs has found similar long-term outcomes for recurrence-free and OS despite a difference in associated lymph node harvest (14 vs. 18) [88].

Rectal NETs represent 20% of GI NETs, although rectal NETs represent only 1–2% of all rectal tumors [89]. The majority are asymptomatic and are found incidentally during colonoscopy. They are frequently located in the mid-rectum [90]. Rectal NETs are graded by mitotic count and KI67 index, as previously described. Rectal NETs are also staged by TNM according to the American Joint Committee on Cancer (AJCC), which incorporates the size and depth of invasion. For lesions 1 cm or smaller that are well differentiated, the risk of lymph node metastasis has been estimated to be 4%, and thus can be removed endoscopically without formal lymphadenectomy [91], and this is supported by NANETS guidelines [92]. For lesions > 2 cm, the risk of lymph node metastasis is 60–80%, so surgical resection by low anterior resection (LAR) or abdominoperineal resection (APR) involving lymphadenectomy is recommended [92,93]. Lesions measuring 1–2 cm is debated among consensus guidelines, with JENTS recommending formal surgical resection for these lesions given that the frequency of lymph node metastasis ranges as high as 18.5–30.4%, even for tumors measuring 1–2 cm [27,34]. ENETS guidelines recommend that lesions 1–2 cm can be resected endoscopically if there is no evidence of muscularis propria invasion or lymph node involvement [93]. NANETS guidelines suggests that if no muscularis invasion or lymph node metastasis is present, transanal local excision is appropriate for lesions measuring 1–2 cm, while low anterior resection with mesorectal excision is suggested for these lesions that do have muscularis invasion or lymph node involvement [92].

## 6. Surgical Management of Appendiceal NETs

NETs of the appendix are commonly found incidentally after appendectomy for appendicitis, found in approximately 1/300 appendectomy specimens [94]. The malignant potential of appendiceal NETs has been found to be primarily affected by tumor size, with tumors < 2 cm rarely being found to have metastatic disease [95], although recent data have questioned this, with metastasis reported in some patients with appendiceal NETs measuring 1–2 cm. Factors associated with metastatic disease in this size range included location at the appendiceal base, lymphovascular invasion, or involvement of the mesoappendix. Thus, right hemicolectomy with lymph node dissection is indicated for lesions > 2 cm, evidence of lymph node involvement, location at the base of the appendix/mesoappendix, or lymphovascular invasion. For lesions 2 cm or smaller localized to the appendix without involvement of the base or mesoappendix, appendectomy is sufficient for the surgical excision of appendiceal NETs [34,96].

## 7. Surgical and Targeted Therapy for Metastatic NETs

The goal of surgical management of metastatic NETs is to control symptoms, reduce tumor burden, and prolong survival. Approximately 12–27% of patients with NETs present with distant metastasis [4,97,98], and patients presenting with metastatic disease have a 4-fold increased risk of death compared to those with local disease [99]. The most common site of metastasis is the liver, and patients with liver metastases may develop symptoms of carcinoid syndrome which include cutaneous flushing attacks, bronchospasm, diarrhea, and carcinoid heart disease [100]. Resection of NET liver metastases not only offers relief from hormonal symptoms and prevents sequelae from carcinoid syndrome but may improve survival, with patients who undergo resection of liver metastases having a 5-year OS rate of 61–74% [52,101,102,103], whereas medical therapy alone has an associated 5-year OS rate of 25–67% [102,103,104], though this comparison of non-randomized studies is subject to significant selection bias. In addition, optimal debulking of 90% or greater of tumor burden also has an associated improvement in symptoms of carcinoid syndrome in comparison to medical management alone (improvement in symptoms in 95% of surgical patients versus 25–80% of patients who underwent medical management) [101,102,105]. Studies investigating the benefit of a lower threshold of tumor debulking, with a goal of 70% or greater tumor burden reduction, found it to still be associated with a reduction in symptoms of carcinoid syndrome or GI obstruction [106], and there was no significant difference in progression-free survival when compared to higher percentages of gross tumor debulking [107,108]. Thus, NANETS guidelines recommend considering surgical debulking when a tumor burden reduction of 70% or greater is possible [109,110].

NETs are unique in the case of metastatic disease in that resection of the primary tumor in the presence of metastasis confers a survival benefit. In a series of 139 patients presenting with symptomatic, liver-metastatic, well-differentiated NET (G1–G2, mitoses: ≤20, Ki-67: ≤20%), primary tumor resection was significantly associated with prolonged survival (median 137 vs. 32 months) [111]. In a series of 314 patients with midgut NETs, 79% of which had liver metastases, and 91% had nodal metastases, it was found that patients whose primary tumor was resected had a median survival of 7.4 years vs. 4 years without primary tumor resection [112]. This benefit was also found in a study of 84 patients with similarly inoperable liver metastases, with a median survival of 159 months with the primary tumor removed versus 47 months without the primary tumor resected [113], and this benefit has been further confirmed in a systematic review that showed a significant overall survival advantage observed in six of eight studies evaluated [114].

In select patients who are ineligible for hepatic tumor debulking, liver transplant may be an option to improve survival in patients with metastatic NETs [115,116]. Per ENETS guidelines and the Milan criteria for liver transplant in the setting of metastatic NET, tumors must be low-grade (G1/G2; Ki-67% < 10) and involve less than 50% of the liver, the primary tumor must have been removed, selected patients must have no extrahepatic disease and have stable disease for 6 months or greater, and must be younger than 55 years of age [117]. The survival advantage of liver transplantation in the setting of metastatic NET is difficult to quantify as there are currently no direct comparison studies between transplantation and cytoreduction. However, a literature review of 705 patients who underwent liver transplantation for metastatic NET reported a 5-year OS of 53% and 5-year DFS of 31% [118] and a 10-year follow-up study evaluating 42 patients who underwent liver transplant for metastatic NET versus 46 who were treated with non-transplant options reported that the 5- and 10-year OS rates were 97.2% and 88.8% for transplant patients versus 50.9% and 22.4% for non-transplant patients [119]. Thus, liver transplant is an option for patients with metastatic NETs.

Non-surgical liver-directed therapies for metastatic NETs include ablation techniques like radiofrequency ablation (RFA) and microwave ablation (MWA), embolization techniques such as transarterial embolization (TAE), transarterial chemoembolization (TACE), and transarterial radioembolization (TARE) (Table 5). Ablation techniques are often used alone or in conjunction with resection for small tumors 4 cm or smaller and up to eight lesions; when performed, these techniques are associated with a 5-year survival rate of 54–84% [106,120,121,122,123,124,125]. RFA is the most performed ablation technique, but in recent years, MWA has become more utilized, with a recent retrospective cohort analysis of 94 patients over 16 years reporting a 5- and 10-year survival probability of 70.2% and 48.2% [126]. Embolization techniques (TAE, TACE, and TARE) utilize the hepatic arterial blood supply to metastatic lesions and are employed more commonly for larger lesions and/or multifocal disease that would not be controlled with ablation techniques. TAE works by blocking the blood supply to the tumor, while TACE combines embolization with localized chemotherapy delivery. TARE uses yttrium-90 (Y-90) microspheres to deliver localized radiation to lesions. The optimal embolization therapy between the three modalities remains a topic of debate; a multi-center retrospective study of 155 patients with NET liver metastases who underwent embolization by one of three methods found TARE to have a higher hazard ratio for OS than TACE in a multivariate analysis, while TAE did not differ significantly from TACE [127].

Though not a liver-directed therapy, PRRT is a targeted treatment option for metastatic NETs that express somatostatin receptors, which works by delivering localized radiation using radiolabeled somatostatin analogs, such as ^177^Lu-DOTATATE. A phase 3 randomized controlled trial (NETTER-1) of 231 patients with well-differentiated, metastatic midgut NETs compared this therapy using ^177^Lu-DOTATATE to long-acting octreotide therapy alone, and PRRT was found to have an improvement in progression-free survival (PFS) (65.2% vs. 10.8% at month 20) [15], though on follow-up analysis, it was determined that this therapy did not improve overall survival in comparison to octreotide alone [128]. However, a follow-up phase 3 trial (NETTER-2) comparing a combination therapy of ^177^Lu-DOTATATE and octreotide vs. an octreotide therapy for gastroenteropancreatic NETs also demonstrated improvement in median progression-free survival by 14 months, with early overall survival data demonstrating no significant difference [129]. These non-surgical directed therapies are important tools in the management of metastatic disease.

## 8. Conclusions

The surgical resection of GI and pancreatic NETs is a cornerstone of treatment, offering curative potential for localized disease and significant benefits in symptom control and possible improvement in survival for metastatic cases. Tailored approaches based on tumor type, location, size, grade, and metastatic involvement are critical, with evidence supporting both aggressive resections and more conservative strategies in select cases. The integration of systemic therapies, targeted modalities, and surgical debulking in metastatic disease highlights the multidisciplinary nature of optimal NET management. Future research and consensus guidelines are essential to refine surgical indications further, especially for intermediate-risk lesions and multifocal presentations, ensuring the best outcomes for patients with this heterogeneous group of malignancies.

## Figures and Tables

**Figure 1 cancers-17-00377-f001:**
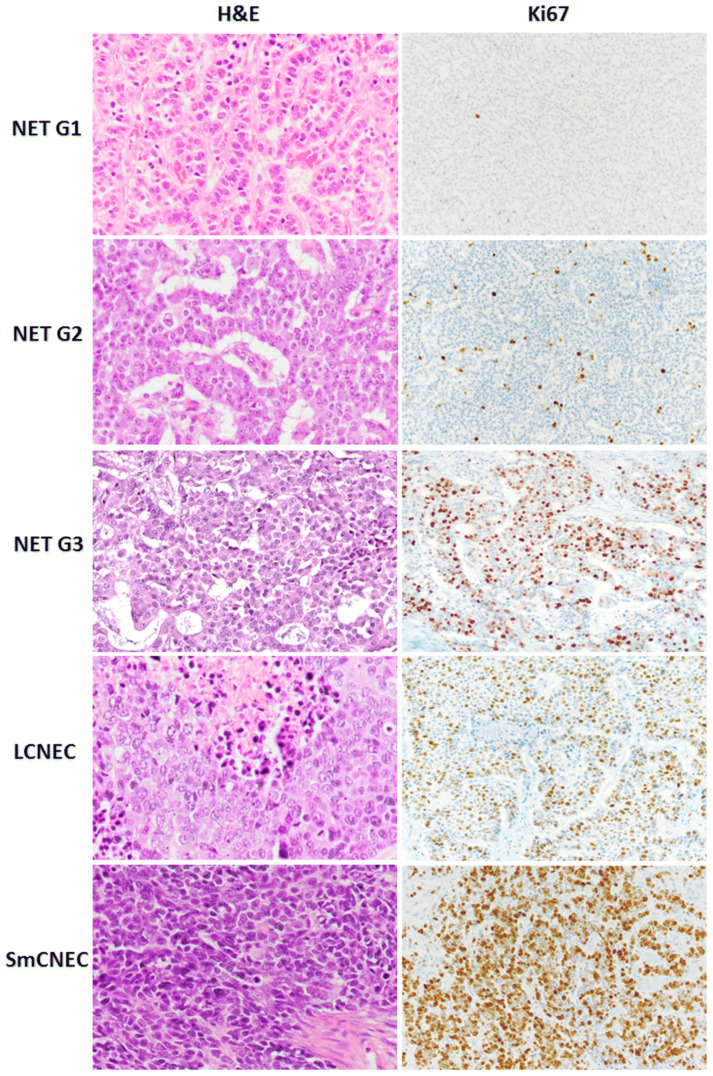
Morphology and Ki67 proliferation index of neuroendocrine tumors and neuroendocrine carcinomas [19].

**Figure 2 cancers-17-00377-f002:**
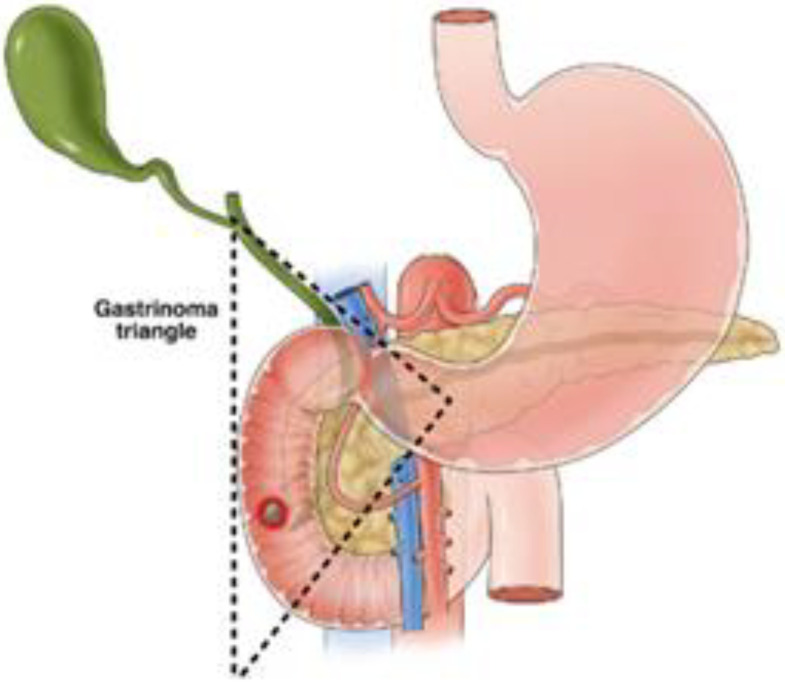
Boundaries of the gastrinoma triangle [71].

**Table 1 cancers-17-00377-t001:** 2022 WHO classification of neuroendocrine tumors.

NET Grade	Ki67 Proliferation Index	Mitotic Index/High Power Field
Grade 1 (G1)	<3%	<2 mitoses/2 mm^2^
Grade 2 (G2)	3–20%	2–20 mitoses/2 mm^2^
Grade 3 (G3)	>20%	>20 mitoses/2 mm^2^
NEC (SmCNEC, LCNEC)	>20%	>20 mitoses/2 mm^2^

**Table 2 cancers-17-00377-t002:** Rindi classification of gastric neuroendocrine tumors and surgical management.

Type	Association	Presentation	Malignant Potential	Management
Type I	Autoimmune atrophic gastritis	Multiple small, well-differentiated lesions	Low	<1 cm: observe/endoscopic resection. 1–2 cm: endoscopic resection or gastrectomy >2 cm or invasion: partial or total gastrectomy
Type II	MEN1, Zollinger–Ellison syndrome	Multiple small, well-differentiated lesions	Moderate (12% metastasis)	<1 cm: observe/endoscopic resection. 1–2 cm: endoscopic resection or gastrectomy >2 cm or invasion: partial or total gastrectomy
Type III	Sporadic, no syndromic association	Solitary, often with gastric ulcer	High (50–100% metastasis)	Partial or total gastrectomy

**Table 3 cancers-17-00377-t003:** Surgical management of small bowel NETs.

Type	Surgical Management
Duodenal NETs	≤1 cm: endoscopic resection (surgical resection + lymphadenectomy if lesion is periampullary) 1–2 cm: endoscopic or surgical resection ≥2 cm or metastasis: surgical resection
Jejunal/Ileal NETs	Surgical resection of primary tumor, lymph nodes, and mesentery including palpation technique of bowel to evaluate for multi-focal disease
Advanced/Multifocal Disease	Cytoreduction, lymphadenectomy, and resection/ablation of hepatic metastases

**Table 4 cancers-17-00377-t004:** Surgical management of pancreatic NETs.

Type	Surgical Management
Insulinomas	Enucleation (>2 mm from the main pancreatic duct) vs. distal pancreatectomy or pancreaticoduodenectomy
Gastrinomas	Distal pancreatectomy or pancreaticoduodenectomy + lymphadenectomy, explore the gastrinoma triangle; MEN1: surgery for lesions > 2 cm (variable guidelines)
Glucagonomas	Distal pancreatectomy or pancreaticoduodenectomy + lymphadenectomy
VIPomas	Distal pancreatectomy or pancreaticoduodenectomy + lymphadenectomy
Somatostatinomas	Pancreaticoduodenectomy, consider enucleation for localized lesions < 2 cm
Non-functional	<2 cm: Observe vs. enucleation (clinical controversy exists) >2 cm: Distal pancreatectomy or pancreaticoduodenectomy + lymphadenectomy

**Table 5 cancers-17-00377-t005:** Liver-directed therapies for metastatic neuroendocrine tumors.

Technique	Mechanism of Action	Indications
Radiofrequency Ablation (RFA)	Thermal energy by alternating radiofrequency current	Small liver tumors (≤4 cm), ≤8 lesions
Microwave Ablation (MWA)	Electromagnetic waves	Alternative to RFA for small liver tumors (≤4 cm)
Histotripsy	Focused ultrasonic waves	No current established indications, evolving field
Transarterial Embolization (TAE)	Blocks the hepatic arterial blood supply, leading to tumor necrosis.	Larger lesions (>4 cm) or multifocal disease unsuitable for ablation techniques
Transarterial Chemoembolization (TACE)	Combines embolization (TAE) with localized delivery of chemotherapy directly to the tumor	Larger lesions (>4 cm), multifocal disease, or unresectable liver metastases requiring additional chemotherapy
Transarterial Radioembolization (TARE)	Delivers targeted radiation via yttrium-90 (Y-90) microspheres to the tumor	Large tumors (>4 cm) or extensive liver involvement unsuitable for TAE/TACE alone

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
