# Peer review of "Surgical Management of Gastroenteropancreatic Neuroendocrine Tumors"

_cancers, 2025, doi:10.3390/cancers17030377_

Round 1

Reviewer 1 Report

Comments and Suggestions for Authors

The review is comprehensive and overall well written. I have some fews suggestions to improve the paper:

1) Some figures or tables (for example summarizing the latest studies in the field) would improve the quality of the manuscript

2) The authors should comment that there are also emerging mini-invasive treatments for pancreatic NETs, for example EUS-guided RFA (in this regard cite the recent study PMID: 36871765)

3) The authors should comment on the impact of primary tumor resection on survival in metastatic well-differentiated NETs (cite the paper PMID: 27956320)

Author Response

1) Some figures or tables (for example summarizing the latest studies in the field) would improve the quality of the manuscript

Response 1: Thank you, additional figures (Figures 1 and 2) and tables (Tables 3-5) summarizing the discussion points of the review have been added.

2) The authors should comment that there are also emerging mini-invasive treatments for pancreatic NETs, for example EUS-guided RFA (in this regard cite the recent study PMID: ) 36871765

Thank you, this was added to the section discussing surgical management of pancreatic NETs.

3) The authors should comment on the impact of primary tumor resection on survival in metastatic well-differentiated NETs (cite the paper PMID: 27956320)

This point is addressed on page 7, paragraph 2. However, this was further expanded on with the addition of the suggested reference PMID:27956320. 

Reviewer 2 Report

Comments and Suggestions for Authors

This is a well-written comprehensive review on the subject. I have some suggestions for improvement:

Many treatment modalities are discussed in this review: endoscopic and surgical resection, radionuclides, different pharmacological options - maybe it is nice to summarise them all in a single diagram/picture.

- Section 1 Maybe a few words on the molecular pathogenesis of NETs should be added. Likewise, just mentioning the systemic medication without more reference to the underlying pathways targeted is also a bit brief, for instance the authors could explain that everolimus act by targeting mTOR signalling (van Veelen et al. The long and winding road to rational treatment of cancer associated with LKB1/AMPK/TSC/mTORC1 signaling. Oncogene. 2011 May 19;30(20):2289-303. doi: 10.1038/onc.2010.630).

-Section 2 - grading. Some pictures showing grades and Ki67 proliferation would enhance presentation and increase value for young residents/clinicians.

The issue of mesenteric ischaemia in NET is not discussed but maybe deserves some attention

Author Response

Comments 1: This is a well-written comprehensive review on the subject. I have some suggestions for improvement:

Many treatment modalities are discussed in this review: endoscopic and surgical resection, radionuclides, different pharmacological options - maybe it is nice to summarise them all in a single diagram/picture.

Reply 1: Thank you for the suggestion. The treatment options and surgical approaches have been summarized in additional tables in the manuscript (Tables 3-5).

- Section 1 Maybe a few words on the molecular pathogenesis of NETs should be added. Likewise, just mentioning the systemic medication without more reference to the underlying pathways targeted is also a bit brief, for instance the authors could explain that everolimus act by targeting mTOR signalling (van Veelen et al. The long and winding road to rational treatment of cancer associated with LKB1/AMPK/TSC/mTORC1 signaling. Oncogene. 2011 May 19;30(20):2289-303. doi: 10.1038/onc.2010.630).

Reply 2: Thank you, A paragraph on the molecular pathogenesis of NETs is added to the Introduction and the mechanisms of the systemic therapies are expanded on, including the suggested reference.

-Section 2 - grading. Some pictures showing grades and Ki67 proliferation would enhance presentation and increase value for young residents/clinicians.

Reply 3: Thank you for the suggestion, Figure 1 has been added to show grades and Ki67 proliferation index.

The issue of mesenteric ischaemia in NET is not discussed but maybe deserves some attention

Reply 4: The possible presenting symptoms of small bowel NETs as mesenteric ischemia is added to line 167.

Reviewer 3 Report

Comments and Suggestions for Authors

The manuscript presents a review of the surgical management of gastroenteropancreatic neuroendocrine tumors. The topic would potentially interest the journal readers. The paper is well-written and well-designed. However, there are a few concerns regarding the current paper:

Major concerns:

Compared with previous reviews addressing the same topic, the current manuscript does not bring any novelty. Thus, the present paper will be of low interest.

The study does not respect the rules for a systematic review.

Minor concerns:

The references should be numbered in their order of appearance in the text. 

Author Response

The manuscript presents a review of the surgical management of gastroenteropancreatic neuroendocrine tumors. The topic would potentially interest the journal readers. The paper is well-written and well-designed. However, there are a few concerns regarding the current paper:

Comment1:

Major concerns:

Compared with previous reviews addressing the same topic, the current manuscript does not bring any novelty. Thus, the present paper will be of low interest.

The study does not respect the rules for a systematic review.

Reply 1: Thank you for the feedback. This review is not intended to be a formal systematic review, and that descriptor was removed from the manuscript.

Minor concerns:

The references should be numbered in their order of appearance in the text. 

Reply 2: Thank you, the reference order has been fixed. 

Round 2

Reviewer 1 Report

Comments and Suggestions for Authors

THe authors improved the manuscript. Thank you!

Reviewer 3 Report

Comments and Suggestions for Authors

The authors made the required changes to the manuscript